# Understanding LAG-3 Signaling

**DOI:** 10.3390/ijms22105282

**Published:** 2021-05-17

**Authors:** Luisa Chocarro, Ester Blanco, Miren Zuazo, Hugo Arasanz, Ana Bocanegra, Leticia Fernández-Rubio, Pilar Morente, Gonzalo Fernández-Hinojal, Miriam Echaide, Maider Garnica, Pablo Ramos, Ruth Vera, Grazyna Kochan, David Escors

**Affiliations:** 1Oncoimmunology Group, Navarrabiomed-Public University of Navarre, IdISNA, 31008 Pamplona, Navarra, Spain; luisa.chocarro.deerauso@navarra.es (L.C.); ester.blanco.palmeiro@navarra.es (E.B.); mzuazo@alumni.unav.es (M.Z.); hugo.arasanz.esteban@navarra.es (H.A.); abocaneg@navarra.es (A.B.); leticia.fernandez.rubio@navarra.es (L.F.-R.); pilar.morente.sancho@navarra.es (P.M.); gfhinojal@gmail.com (G.F.-H.); mechaidg@navarra.es (M.E.); mgarnics@navarra.es (M.G.); pramosca@navarra.es (P.R.); 2Department of Medical Oncology, Complejo Hospitalario de Navarra CHN-IdISNA, 31008 Pamplona, Navarra, Spain; ruth.vera.garcia@navarra.es

**Keywords:** LAG-3, immune checkpoint, cancer signaling, immunotherapy, targeted therapy

## Abstract

Lymphocyte activation gene 3 (LAG-3) is a cell surface inhibitory receptor with multiple biological activities over T cell activation and effector functions. LAG-3 plays a regulatory role in immunity and emerged some time ago as an inhibitory immune checkpoint molecule comparable to PD-1 and CTLA-4 and a potential target for enhancing anti-cancer immune responses. LAG-3 is the third inhibitory receptor to be exploited in human anti-cancer immunotherapies, and it is considered a potential next-generation cancer immunotherapy target in human therapy, right next to PD-1 and CTLA-4. Unlike PD-1 and CTLA-4, the exact mechanisms of action of LAG-3 and its relationship with other immune checkpoint molecules remain poorly understood. This is partly caused by the presence of non-conventional signaling motifs in its intracellular domain that are different from other conventional immunoregulatory signaling motifs but with similar inhibitory activities. Here we summarize the current understanding of LAG-3 signaling and its role in LAG-3 functions, from its mechanisms of action to clinical applications.

## 1. Molecular Characterization of LAG-3

### 1.1. Function

LAG-3 (CD223) is an inhibitory receptor first described in *in vitro*-activated T cells. LAG-3 has multiple biological effects on the function of T cells and CD4 T cell activation, usually inhibitory [1,2]. LAG-3 does not seem to inhibit T cell function in the absence of CD4 activation but may interfere with CD4 coreceptor function. LAG-3 negatively regulates proliferation, activation, effector function, and homeostasis of both CD8 and CD4 T cells, as shown in LAG-3 knockout mice and by LAG-3 blockade with antibodies in human cells [1,3,4,5]. Thus, the evidence suggests that LAG-3 interferes with a common pathway to both CD4 and CD8 activation and regulates the activation and expansion of T memory cells [6].

LAG-3 plays a regulatory role in the immune system comparable to PD-1 and CTLA-4, generally consisting of inhibition of cell proliferation, immune function, cytokine secretion, and homeostasis [1,2,3,7]. LAG-3 can be upregulated under various antigen stimulation conditions [3,8,9]. Its expression is induced by TCR stimulation or cytokine stimulation, and LAG-3 is upregulated within activated, cytokine-expressing T cells [9,10,11,12]. LAG-3 associates with the TCR:CD3 complex at the T cell membrane and negatively regulates TCR signal transduction, which in turn terminates cell proliferation and cytokine secretion in response to CD3 signaling [13]. LAG-3 and CD3 co-engagement in the immunological synapse is necessary to down-modulate TCR signal transduction. Among other consequences, the simultaneous engagement of LAG-3/TCR with their ligands inhibits TCR:CD3-dependent intracellular calcium fluxes. Similarly to PD-1, constitutive LAG-3 expression is frequently associated with exhausted T cells, and it is generally regarded as an exhaustion marker for CD4 and CD8 T cells in response to repetitive antigen stimulation in cancer and chronic viral infections [14,15,16,17,18,19]. Indeed, LAG-3 has been found to physically associate with TCR in both CD8 and CD4 T cells following TCR engagement, downregulating TCR-dependent signaling cascades and thereby dampening T cell responses [13] (Figure 1).

### 1.2. Genetic Structure of the LAG-3 Gene and Domain Organization

Back in 1990, Dr. Frédéric Triebel and colleagues identified *lag-3* as a novel lymphocyte activation gene closely related to CD4. The authors found a 2-kb mRNA to be selectively transcribed in activated human T and NK cells [8]. *Lag-3* encodes a protein with a molecular weight of 70 kDa and resides on the distal part of the short arm of chromosome 12 (12p13.32) in humans and in chromosome 6 in mouse. The *lag-3* gene encodes a 498-amino acid type I membrane protein [20]. Its locus is adjacent to that of the CD4 co-receptor and has a similar sequence and intron/exon organization, strongly indicating that *lag-3* and *CD4* genes have evolved by gene duplication from a pre-existing common evolutionary ancestor gene encoding a two-IgSF-domain structure.

The LAG-3 protein structure can be divided into extracellular, transmembrane, and intracellular regions (Figure 2). The gene has 8 exons. Exon I encodes 19 aa of the hydrophobic leader peptide; exon II encodes 9 aa of the leader peptide (9 aa) and 41 aa of the extracellular region (41 aa); the rest of the extracellular region is encoded by exon III (101 aa), IV (90 aa), V (92 aa), and VI (81 aa); exon VII encodes the transmembrane region (44 aa); and exon VIII (21 aa), the highly charged cytoplasmic region [8].

The LAG-3 protein corresponds to a mature protein of 525 residues with a molecular mass of approximately 50 kDa. Both CD4 and LAG-3 extracellular regions are composed of four extracellular immunoglobulin superfamily-like (IgSF) domains (D1-D4) (Figure 2). D1 is an Ig variable-like region (V-SET type), and it includes an extra, unique, proline-rich loop (30 residues) in the middle of the domain (between the C and C′ β strands of the D1 domain) and an unusual intra-chain di-sulfide bridge [8]. Unlike CD4, this loop mediates LAG-3–MHCII interactions [21]. Domains D2, D3, and D4 of LAG-3 belong to the C2-SET, while the CD4 D3 domain is of a V-SET type. D1 and D3 of LAG-3, as well as D2 and D4, share strong internal structural homologies. However, CD4 and LAG-3 only share less than 20% amino acid sequence homology. A β-strand between D1 and D2 and between D3 and D4 extends rigidly between these substructures [20].

Additionally, compared to CD4, the LAG-3 protein has a longer connecting peptide between the membrane-proximal D4 domain and the transmembrane region. LAG-3 is cleaved within this connecting peptide, resulting in the release of a soluble form [21]. This cleavage has been described to be mediated by two transmembrane metalloproteases, ADAM10 and ADAM17, regulated through two distinct mechanisms. While ADAM10 mediates constitutive LAG-3 cleavage, ADAM17 mediates LAG-3 cleavage induced by TCR signaling in a PKCθ-dependent manner [22]. Interestingly, there is evidence that shedding of LAG-3 from the cell surface enhances T-cell proliferation and effector function, which adds another activating level of immune regulation by LAG-3 [21,22].

The cytoplasmic tail of LAG-3 mediates intracellular negative signal transduction, as its deletion completely abrogates its inhibitory function. The cytoplasmic tail contains three conserved motifs (Figure 2): (1) a potentially phosphorylatable serine (S484), (2) a KIEELE motif, and (3) a glutamate-proline dipeptide multiple repeats motif (EP motif). Nevertheless, there is conflicting evidence on how these motifs affect LAG-3 molecular function and downstream signaling. The potential serine phosphorylation motif (S484) has been described within the FxxL motif, although no specific function has been ascribed to it other than its correlation with IL-2 production. This serine may be analogous to the protein kinase C binding site in the CD4 molecule [23].

The KIEELE motif corresponds to a highly conserved short sequence not found in other proteins, and its relevance to LAG-3 function remains controversial. According to Workman et al., the deletion of KIEELE completely abrogates LAG-3 function on CD4 T cells, highlighting its importance for LAG-3 inhibitory signaling [3,24]. In addition, an alanine scanning mutagenesis of KIEELE residues showed that lysine residue (K468) was absolutely required for LAG-3 downstream signaling, with minor contributions by E470 and E471 in CD4 function. However, according to Maeda and colleagues, LAG-3 mediates a cell-intrinsic, negative inhibitory signal independently of KIEELE via two distinct mechanisms that are dependent on the FxxL motif in the membrane proximal region and the carboxy-terminal EP repeats [25].

The EP repeat motif is present in some proteins in diverse species and cellular locations, which could be an indicator of a potentially common biological function. For example, PDGF-R, LckBP1, and SPY75 contain similar EP regions that contribute to their signaling mechanisms. It was proposed that the EP motif was key in counteracting the CD3/TCR activation pathway via association to LAP protein (LAG-3 associated protein), allowing LAG-3 co-localization with CD3, CD4, and CD8 within lipid rafts [26]. According to Workman and colleagues, the deletion of the EP motif or the S484A mutation maintained LAG-3 activity and function in CD4^+^ and CD4^−^ T cells, suggesting that these two features may not have been essential [3]. Nevertheless, it has to be stressed that the inhibitory activity of LAG-3 was only completely abrogated in CD4^−^ T cells if both KIEELE and EP motifs were removed. The authors proposed that the EP motif could play a role in preventing LAG-3 from acting as a stimulatory co-receptor such as CD4 rather than cooperating with KIEELE in inhibitory signal transduction.

As LAG-3 is localized in lipid rafts following antigen stimulation [26], its cytoplasmic domain could be mediating its location in the cell membrane. LAG-3 co-localizes with glycosphingolipid-enriched microdomain complexes containing CD3, CD4, or CD8. After TCR engagement, LAG-3 co-localizes with CD3 and ganglioside GM1 in the immunological synapse. However, no systematic follow-up studies on these motifs and their functions have been reported to date. The structural nature of LAG-3 needs to be elucidated to understand its mechanisms of action on the regulation of T cell activation.

Apart from its surface expression, LAG-3 can also be stored in lysosomes, and it is translocated to the membranes of activated T cells via its cytoplasmic domain by protein kinase C signaling [27].

### 1.3. LAG-3 Ligands

MHC-II molecules are considered the canonical ligand of LAG-3, with which it stably interacts through the D1 domain with a significantly higher affinity than with CD4 [20]. This association negatively regulates T cell activation, cytotoxicity, and cytokine production [28]. In fact, LAG-3-Ig fusion proteins act as competitors in CD4/MHC class II-dependent cellular adhesion assays [29]. Once LAG-3 is bound to MHC-II, it transmits inhibitory signals via its cytoplasmic domain and inhibits CD4 T cell activation [1,2]. Nevertheless, the molecular mechanisms of signal transduction remain broadly unknown compared to other immune checkpoint molecules. LAG-3:MHC class II binding contributes to tumor escape from apoptosis [30] and facilitates recruitment of tumor-specific CD4 T cells, but with a reduction of the CD8 T cell response [31].

Initially, the high-affinity LAG-3:MHC class II interaction was proposed to be the main mechanism of the inhibitory activities of LAG-3 through competition with CD4:MHC II binding [1,2]. However, it still remains controversial whether the interaction with MHC-II is solely responsible for mediating LAG-3 immunosuppressive functions, in light of the recent identification of additional ligands.

The second LAG-3 ligand to be described was galectin-3 (Gal-3), a 31 kDa galactose-binding lectin that modulates T cell activation. Gal-3 has been shown to be highly expressed in various tumor cells and in activated T lymphocytes [32,33,34,35]. Gal-3 binds to LAG-3 and appears to be required for optimal inhibition of CD8 T cell cytotoxic function. Gal-3 shapes antitumor-specific immune responses by suppressing activated antigen-committed CD8 T cells via LAG-3 expression in the tumor microenvironment and inhibiting expansion of plasmacytoid dendritic cells [35].

The liver-secreted protein fibrinogen-like protein 1 (FGL1) was recently identified as a new LAG-3 functional ligand [36]. FGL1 is a member of the fibrinogen family and possesses immunosuppressive activities by inhibiting antigen-specific T cells through LAG-3 binding. FGL1 interacts with LAG-3 via the fibrinogen-like domain with the D1 and D2 Ig-like domains of LAG-3. Its expression is induced by IL-6 and is highly up-regulated in several human cancers such us lung cancer, prostate cancer, melanoma, and colorectal cancer. Indeed, it was demonstrated that elevated plasma FGL1 levels were associated with poor cancer outcomes and anti-PD-1 therapy resistance [36]. Thus, this binding was described as a novel mechanism of immune evasion, and its blockade potentiated anti-tumor T cell immunity in pre-clinical models. The FGL1/LAG-3 blockade may synergize with anti-PD-1/anti-PD-L1 therapy. A single point mutation (Y73F) in the D1 C’ strand domain of LAG-3 disrupted LAG-3/MHC-II binding [3,4,20], but not FGL1-Ig binding. Nevertheless, this mutant LAG-3 could still inhibit T cells, demonstrating that FGL1/LAG-3 interactions correspond to a tumor immune evasion mechanism which is non-redundant to MHC-II binding.

Taken together, these data suggest that LAG-3 interactions with several ligands are important for its inhibitory function, and unlike what was thought before, LAG-3 does not function primarily by disrupting CD4:MHC-II interactions.

## 2. Regulation of LAG-3 Expression

### 2.1. Cell and Tissue Expression

LAG-3 expression has been characterized by multiple studies in murine cells, in human blood cells, and other tissues (Figure 3). LAG-3 is expressed by different T cell subsets such as activated CD4 T helper (Th) and cytotoxic CD8 T cells (CTL) [37]. LAG-3 is expressed by most of the activated CD4 differentiation subsets (Th1, Th0) with the exception of Th2. T cell activation seems to be a requirement for LAG-3 expression on the cell surface [27]. LAG-3 expression is also upregulated by cytokines such as IL-2 and IL-12 on activated T cells [10,38,39], and its expression correlates with increased IL-10 production [40]. Surface LAG-3 protein expression and LAG-3 shedding by activated CD4 human T cells strongly correlates with IFN-γ production, a potent MHC-II inducer. In addition, LAG-3 expression inversely correlates with IL-4 production [10]. LAG-3 co-localizes with CD3 and CD4/CD8 clusters after its translocation to the membrane through lipid rafts [6]. Tumor-infiltrating, antigen-specific CD8 T cells have been reported to be negatively regulated by LAG-3 in human ovarian cancer, and PD-1/LAG-3 co-expression in peripheral blood T cells is a biomarker of strong T cell dysfunctionality in lung cancer [41,42].

LAG-3 has also been described as being expressed by subsets of natural killer cells (NK) and invariant NK T cells. Indeed, the *lag-3* locus is close to the NK gene complex, which is expressed on activated NK cells. However, LAG-3 does not participate in a specific mode of natural killing in human NK cells [8,9,43].

LAG-3 is expressed in activated effector CD4 T cells [44], and particularly in activated natural and inducible regulatory T cells (nTregs and iTregs, respectively). Foxp3^-^CD25^low^ IL-10^+^ type 1 regulatory T cells (Tr1) also highly express LAG-3 [45,46], and its co-expression with CD49b^+^ has been used as an identifying Tr1 feature [45,46,47]. In fact, it has been shown that the IL-27/LAG-3 axis enhances Treg suppressive function [48]. Overall, the evidence supports the requirement of LAG-3 expression for maximal suppressing functions of natural and inducible Tregs [13,49].

LAG-3 expression has also been described in exhausted CD8 PD1^+^ tumor-infiltrating lymphocytes (TILs) [44,45,50]. However, its role in immune escape is still controversial, as it has been described that LAG-3 expression in TILs is related to a positive prognosis in esophageal cancer, non-small cell lung cancer, Hodgkin’s lymphoma, and microsatellite instability-high colorectal cancer cases [51,52,53,54]. Expression of LAG-3 by TILs is related to suppression of antigen-specific CD8 T cell function [51]. In addition, LAG-3 co-expression with other immune checkpoint molecules such as CD274 and IDO1 in TILs serve as a prognostic biomarker for cancer prognosis [53].

LAG-3 expression on activated B cells has been described to be T cell-dependent [55]. High expression of LAG-3 has also been shown in chronic lymphocytic leukemia cells (CLL), characterized by clonal expansion of mature B-cells (CD5^+^ CD23^+^ CD27^+^ Ig^low^). LAG-3 is also expressed by a natural regulatory plasma cell subset (LAG-3^+^CD138^hi^ plasma cells, or Bregs), differentiated in a B cell receptor (BCR)-dependent manner [56]. Interestingly, this cell subset exhibits a distinct plasma cell-specific epigenomic/transcriptional signature and molecular program. In this case, bruton tyrosine kinase (Btk) and BCR signaling control the development of LAG-3^+^ Bregs independently of TLR signaling and T cell help. In contrast, Khsheibun and collaborators found that LAG-3 mRNA and protein expression in Epstein–Barr virus transformed B cells without T cell help [57].

There is evidence that LAG-3 can also be expressed by other cells not belonging to the lymphocytic lineage. This is the case in LAG-3 expression by plasmacytoid dendritic cells (pDCs). LAG-3 regulates CD11c^low^ B220^+^ PDCA-1^+^ pDC cell homeostasis in a selective cell-intrinsic and cell-extrinsic manner. It has been shown that activated pDCs can even secrete about 5 times more soluble LAG-3 than activated T cells [14]. Although it is not expressed in more DC subsets, soluble LAG-3 causes effective phenotypical and functional maturation and activation of human monocyte-derived dendritic cells (DC) in the presence of GM-CSF and IL-4 [58]. In addition, LAG-3-induced matured DCs secrete IL-12 and TNF-α inflammatory cytokines and possess strong allostimulatory capacities. In fact, a LAG-3 fused to an immunoglobulin module has been used as a potent vaccine adjuvant due to its potent capacity to cause full maturation of human monocyte-derived DC.

Interestingly, LAG-3 has also been shown to be expressed in neurons, where it specifically binds to and co-endocytoses with pathologic α-synuclein [59,60,61].

### 2.2. Genetic and Epigenetic Regulation of LAG-3 Expression

Transcription from the *lag-3* gene produces 4 transcripts or splice variants (*lag*3-201, *lag*3-202, *lag*3-203, *lag*3-204) (Table 1). *Lag*-3 genes can be classified into 190 orthologues (26 in primates, 30 in rodents, 33 in Laurasiatheria, 92 in all placental mammals, 14 in Sauropsida,) and 10 paralogues (*IL18RAP, SIGIRR, IL1R2, IL1RL1, IL1RAPL2, IL18R1, IL1RAPL1, IL1RAP, IL1R1, IL1RL2*). *Lag*-3 is a member of the 1 Ensembl protein family (PTHR11890 (the interleukin 1 receptor) and its precursor family) and of the Human CCDS set (CCDS8561.1). The *lag*-3 gene maps to 6,881,678-6,887,621 in GRCh37 coordinates (CM000674.1), and its locus corresponds to chromosome 12: 6,772,512-6,778,455 in the forward strand. The LAG-3 protein is assigned to the P18627 UniProtKB identifier.

The *lag-*3 gene expresses two canonical transcripts with the same transcription start site. Transcript LAG3-202 results in a truncated version of the full-length protein as a consequence of not having the last three exons. An extended promoter is also predicted. Binding sites for the transcriptional repressor CCCTC-binding factor are predicted (Figure 4).

The epigenetic regulation of *Lag-*3 transcription is largely unknown and might involve DNA methylation [63]. Methylation has been shown to regulate the expression of PD-1, PD-L1, and CTLA-4 in various cancers. For example, methylation of the PD-L1 promoter predicts survival in colorectal cancers, head and neck squamous cell carcinoma, and acute myeloid leukemia [64,65,66]; PD-1-promoter methylation is associated with recurrence-free survival in prostate cancer [65]; and CTLA-4 methylation correlates to response to anti-PD-1 and anti-CTLA-4 immunotherapy in melanoma [67]. For instance, a TP53-associated immune prognostic signature (TIPS) is positively correlated with high expression of LAG-3 and other critical immune checkpoint biomarkers in the TCGA cohort [68].

CpG islands in the *Lag-*3 promoter have been described as highly hypomethylated. Histone 3 lysine 9 (H3K9) and lysine 27 (H3K27) were shown to be trimethylated in human primary breast tumor tissues compared with normal tissues [69]. This was associated with low enrichment of repressive histones in the promoter region. In addition, DNA methylation of the *Lag-*3 gene correlated with LAG-3 expression by tumor and immune cells, immune cell infiltration, transcriptional activity, and overall survival in clear cell renal cell carcinoma. These results suggested that *Lag-*3 expression was likely epigenetically regulated via methylation [70]. *Lag-*3 expression has been described to be negatively regulated by microRNA-146 in cutaneous T cell lymphoma [71]. In addition, class-specific or class-selective histone deacetylase 6 (HDAC6) inhibition downregulated the expression of LAG-3 in T cells from melanoma patients, thus alleviating T cell suppression [72]. Similar effects were reported on TIM-3, PD-1, and PD-L1 expression.

The transcription factor Egr-2 has been shown to control the activities of CD4 CD25^-^ LAG-3^+^ regulatory T cells, which may be also involved in *Lag-*3 transcription. Furthermore, IL-10-secreting Egr-2^+^ LAG-3^+^ CD4 Tregs could be key for the control of peripheral immunity [73]. Moreover, DHEA-Box Helicase 37 (DHX37) expression has been recently correlated with *Lag-*3 by gene correlation analyses [74].

There is strong evidence showing that LAG-3 protein expression is related to an “epigenetic exhaustion state” in T cells [75]. From these studies, it can be concluded that DNA and histone modifications are certainly involved in LAG-3 upregulation in cancer, suggesting that epigenetic modifications could be useful as a potential predictive biomarker of response to immune checkpoint blockade immunotherapies. Nevertheless, further studies are needed to clarify the role of methylation in dysfunctional cell immunophenotypes in cancer and to define its role as a potential diagnostic/prognostic biomarker and therapeutic target.

## 3. LAG-3 in Disease

### 3.1. Cancer

LAG-3 expression in T cells is generally assumed to be a marker for aggressive progression in a broad spectrum of different human tumors such as melanoma, Hodgkin’s lymphoma, chronic lymphocytic leukemia, colorectal cancer, ovarian cancer, hepatocellular carcinoma, renal cell carcinoma, gastric cancer, follicular lymphoma, prostate cancer, head and neck squamous cell carcinoma, non-small cell lung cancer, malignant pleural mesothelioma, breast cancer, anal squamous cell carcinoma, pancreatic cancer, oropharyngeal squamous cell carcinoma, and urothelial carcinoma, among others [30,41,42,51,76,77,78,79,80,81,82,83,84,85,86,87]. Its expression has also been described as a molecular factor driving T cell exhaustion in several cancers such as pancreatic ductal adenocarcinoma [80]. LAG-3-expressing tumors are usually associated with poor survival, although a high expression of LAG-3 is associated with favorable overall survival and disease-free survival in some solid tumors, particularly in early-stages, such as breast cancer, ENKTL nasal type, NSCLC, and TNBC [88].

The past 5 years have seen an improved understanding of the mechanisms underlying resistance to checkpoint inhibition such as upregulation or activation of the alternative immune checkpoints including PD-1, CTLA-4, and LAG-3. LAG-3 and PD-1 co-expression is associated with poor disease prognosis in several malignancies. Its upregulation in tumor antigen-specific T cells implies impaired cell-mediated immunity and marks T cells with proliferative dysfunctionality [42,89]. The prognostic significance of immune checkpoint expression in the microenvironment of primary and metastatic cancers is now a reality.

Cancer patients with high expression of LAG-3 in T lymphocytes show activated pro-apoptotic phenotypes in T cells, which is associated with increased PD-1 expression and poor survival after a PD-1 blockade [90]. PD-1/LAG-3 expression is a marker for exhausted T lymphocytes infiltrating tumors [17], which a high-risk signature, and a clinical factor predictive of overall survival [91,92]. NSCLC patients who do not respond to immunotherapy present highly dysfunctional systemic T lymphocytes that simultaneously express PD-1 and LAG-3 following TCR stimulation. These dysfunctional T lymphocytes are resistant to anti-PD-L1/PD-1 monotherapies, possibly due to the expression of LAG-3 [42]. It has been shown in different experimental models that LAG-3 and PD-1 are co-expressed in TILs from progressing tumors, and that their co-blockade has synergistic effects against immune escape, increased antitumor response, enhanced T cell proliferation, and cytokine production [41,42,56,93,94]. Indeed, the blockade of PD-1/LAG-3 pathways during T-cell priming increases T cell proliferation, cytokine production, and antitumor functions [41]. Overall, these studies suggest that tumors in which immune evasion is predominantly mediated by LAG-3 are less sensitive to a PD-1 blockade. Hence, this opens the possibility of eventually using LAG-3 as a stratifying biomarker in immunotherapy. Several clinical trials are currently ongoing, investigating a combinatory blockade of immune checkpoints that include LAG-3.

### 3.2. Parkinson’s Disease

Emerging preclinical and clinical evidence suggests that LAG-3 is associated with an increased risk of Parkinson’s disease (PD) [59]. LAG-3 is expressed in neurons and acts as a binding receptor for the pathologic fibrillary α-synuclein, mediating its aggregation, intracellular transmission, and spreading [60]. This process involves pre-formed α-synuclein fibrils followed by exogenous endocytosis by LAG-3 engagement on neurons [61]. Some LAG-3 single-nucleotide polymorphism variants have been correlated with increased PD risk, especially in the female population. Soluble LAG-3 in serum and cerebrospinal fluid (CSF) have been shown to associate to PD clinical development and negative progression. Thus, a disruption of the immune homeostasis caused by LAG-3 dysfunction in the central nervous system could initiate neuron-to-neuron α-synuclein aggregation and PD progression. In conclusion, LAG-3 could serve as a possible therapeutic target to slow the progression of α-synucleinopathies.

### 3.3. Cardiovascular Diseases

LAG-3 protein expression has been shown to correlate with increased coronary heart disease (CHD) and increased myocardial infarction (MI). LAG-3 accumulates in cardiac allografts undergoing rejection episodes to fully vascularized, heterotopic, allogeneic heart transplantation [95]. LAG-3 deficiency has also been associated in clinical studies with increased risk of coronary artery disease (CAD) due to Tr1 dysfunction [96].

### 3.4. HDL Hypercholesterolemia

LAG-3 protein expression levels have been significantly associated via transcriptomic studies with high HDL cholesterol (HDL-C) and HALP (HDL-C ≥ 60 mg/dL). Indeed, LAG-3 deficiency altered lipid raft formations and cell phosphosignaling, processes leading to an enhanced proinflammatory state, and increased production of inflammatory cytokines such as TNFα [97,98].

### 3.5. Inflammatory Bowel Disease

LAG-3 has been reported to be a modulator of T cell regulation in inflammatory responses in the intestine [48]. In addition, LAG-3^+^-regulatory T cells are required to suppress the inflammatory activities of CX3CR1^+^ macrophages to maintain tissue homeostasis during lymphoid cell-driven colitis [99]. Moreover, LAG-3^+^ cells have been shown to be increased in the inflamed mucosa and correlate with endoscopic severity and disease phenotype in ulcerative colitis [12].

### 3.6. Multiple Sclerosis

Germline allelic variation of the LAG-3 gene has been described to confer susceptibility to multiple sclerosis (MS). Particularly, at least three single-nucleotide polymorphism (SNPs) have been significantly (*p* < 0.05) associated with increased risk of MS, possibly by derangement of immune homeostasis [100]. In this study, three distributions of SNPs were found to be significant for MS susceptibility: rs19922452, rs951818, and rs870849. Rs870849 represents LAG-3 with an amino acid substitution of a nonpolar isoleucine for an uncharged polar threonine (Thr455Ile) that may alter protein function and conformation. SNPs rs19922452 and rs951818 are located in a noncoding region, so their implication in increased risk is unclear. However, this remains to be determined through functional studies of the LAG-3 protein to validate the importance of these results and to identify protective SNP haplotypes.

### 3.7. Diabetes Mellitus

LAG-3 expression could effectively prevent some autoimmune disorders, as it has been described as limiting the pathogenic potential of CD4 and CD8 T cells in the initiation phase and the driving phase of diabetes onset [101]. In fact, the absence of LAG-3 accelerates autoimmune diabetes [101]. The deletion of LAG-3 resulted in faster autoimmune diabetes development in both male and female, non-obese diabetic (NOD) mice, a model for type 1 diabetes (T1D) in humans. The absence of LAG-3 led to a faster accumulation of T cell infiltration within NOD pancreatic islets. Even a partial reduction in LAG-3 expression led to rapid diabetes disease development. LAG-3 is also important to inhibit Th1 cell activation, reducing T-cell autoreactiveness and T1D [102]. In this case, the therapeutic enhancement of LAG-3 functions could serve as a treatment for T1D.

### 3.8. LAG-3 in Infection

LAG-3 expression has also been linked to increased pathology in certain infections. High accumulation of LAG-3^+^ CD138^hi^ natural regulatory plasma cells was associated with impaired control of immunity against *Salmonella* infection, leading to higher bacterial load and a reduced survival rate [56]. Infection with *Plasmodium* parasites (*P. yoelii 17XL, P yoelii 17XNL, P. chabaudi, P. vinckei*, and P. *berghei*) causes increased PD-1 and LAG-3 expression by activated CD4 cells [103]. In fact, CD11a^hi^ CD49d^hi^ CD4 T cells express PD-1 and LAG-3 during infection with *Plasmodium* parasites. Interestingly, the in vitro blockade of PD-1/LAG-3 interactions enhanced cytokine production in response to infection. LAG-3 was also shown to be expressed in CD4 T cells and NK cells during active *Mycobacterium tuberculosis* infections, enhancing high bacterial burdens together with changes in Th1 responses [104]. Accordingly, LAG-3 expression was shown to be increased in the lungs, and particularly within the granulomatous lesions. LAG-3 could also be relevant in human immunodeficiency virus (HIV) infection, as its upregulation is associated with a high viral load within a T cell exhausted subset, correlating with faster disease progression [105]. Nonpathogenic simian immunodeficiency virus (SIV) primary infections induce DNA methylation in the LAG-3 coding gene. A similar mechanism may be occurring in HIV-infected patients, which could contribute to persisting metabolic and inflammation disorders [106].

LAG-3 expression is significantly up-regulated in hepatitis B virus (HBV)-specific CD8 T cells, acting as a suppressor of HBV-specific, cell-mediated immunity or even to the pathogenesis of hepatocellular carcinoma [84]. LAG-3 expression was also correlated with human papillomavirus (HPV) status [81]. Thus, LAG-3 inhibitors may help the immune system, overcoming immune exhaustion to fight viral infections. LAG-3 is also expressed on CD8 T cells during chronic hepatitis C virus (HCV) infection, inhibiting cell proliferation, cytotoxicity function, and cytokine production. LAG-3^+^ CD4 T cell numbers were negatively associated with HCV-neutralizing antibody responses [87]. Exhausted LAG-3^+^ CD8 T cells were also associated with other chronic viral infections [107,108]. In agreement with this, lymphocytic choriomeningitis viral (LCMV) and herpes simplex virus 1 (HSV-1) infections also showed an increase in LAG-3 expression after infection [14,109,110,111,112].

## 4. Clinical Landscape of LAG-3-Targeted Therapy

Cancer immunotherapies, including immune checkpoint inhibitor blockades, stimulate the immune system to recognize and eliminate tumor cells and could also be important for enhancing responses toward infectious agents. These therapies act mainly on the reactivation of T lymphocytes. However, not all patients respond to these therapies, which represents a significant clinical, economical, and ethical problem [113,114,115,116]. The inhibitory co-receptors that modulate T cell activation during antigen presentation are highly important immune checkpoints. These molecules are generally present in the immune synapse, together with the TCR, and modulate its signal transduction capacities. Some examples are PD-1, CD244, CD160, TIM-3, CTLA-4, and LAG-3 [117]. Particularly, ICIs blocking PD-1/PD-L1 interactions have revolutionized oncology, with very promising results for cancer treatment [118,119,120,121]. Due to the efficacy of these ICIs, a set of second-generation ICIs such as anti-LAG-3 antibodies and combinations are being evaluated at the preclinical and clinical levels [7].

LAG-3 was first exploited in clinical trials in 2006 as a LAG-3-Ig fusion protein to take advantage of its immune-stimulating activities as a soluble protein (Eftilagimod alpha, IMP321). Nowadays, there are several LAG-3-antagonistic immunotherapeutics models at various stages of clinical and pre-clinical development (Table 2). In addition, combinations blocking LAG-3 together with other immune checkpoints (PD-1, PD-L1, TIM-3, CTLA4) are also being studied [108,122]. A new generation of bispecific PD-1/LAG-3 blocking agents have shown strong capacities to specifically target PD-1^+^ LAG-3^+^ highly dysfunctional T cells and enhance their proliferation and effector activities [123].

## 5. Conclusions

The lymphocyte-activating gene-3 (LAG-3) is a key regulator of immune homeostasis as a next-generation immune checkpoint. Several LAG-3 blockade immunotherapeutic models are being pursued at various stages of clinical and pre-clinical development. However, the mechanisms leading to LAG-3 inhibitory functions are still poorly understood. This is a major clinical problem and one of the most significant challenges for drug development in immunooncology. The understanding of the mechanisms of PD-1/LAG-3 co-signaling in driving T cell dysfunctionality could help us understand the basis for resistance to immunotherapy and uncover mechanisms to counteract the resistance and benefit from immunotherapies. Indeed, the co-blockade of LAG-3 with PD-1 is showing encouraging results, as recently announced for the treatment of metastatic melanoma in the RELATIVITY-047 phase 3 clinical trial (https://clinicaltrials.gov/ct2/show/NCT03470922, accessed on 2 May 2021). The combination of relatlimab and nivolumab met the primary endpoint of progression-free survival.

Concluding, a deeper understanding of the basic mechanisms underlying LAG-3 intracellular signaling will provide insight for further development of novel strategies for cancer, infections, autoimmune disorders, and neurological diseases.

## Figures and Tables

**Figure 1 ijms-22-05282-f001:**
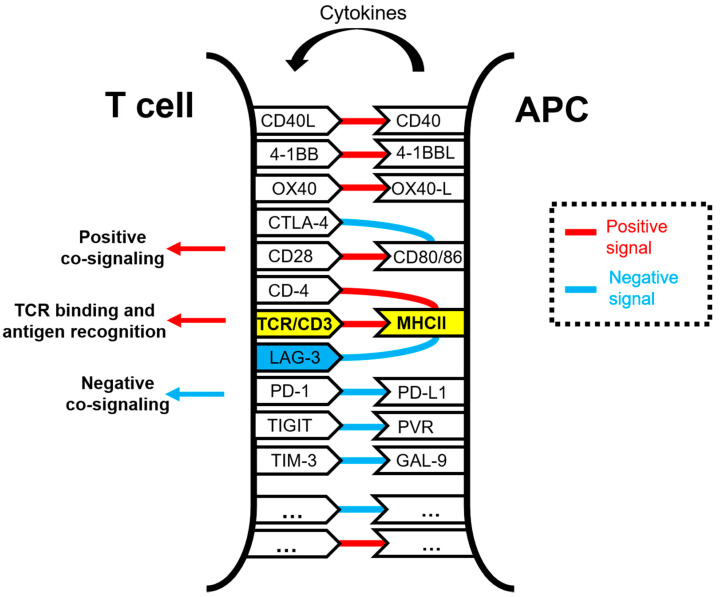
Schematic representation of the molecular interactions occurring within the immunological synapse between a T cell and an antigen-presenting cell (APC) during antigen presentation and T cell activation. The TCR/CD3 and MHC complexes are highlighted in yellow. Some of the well-known co-stimulatory and co-inhibitory receptor–ligand interactions are shown in the figure, linked by red lines for activating interactions and with blue lines for inhibitory interactions. Antigenic peptides presented by APCs loaded in MHC molecules are specifically recognized by the TCR, while other interactions take place simultaneously to deliver co-stimulatory and co-inhibitory signals. These interactions are integrated within T cells. The LAG-3 molecule is highlighted in blue.

**Figure 2 ijms-22-05282-f002:**
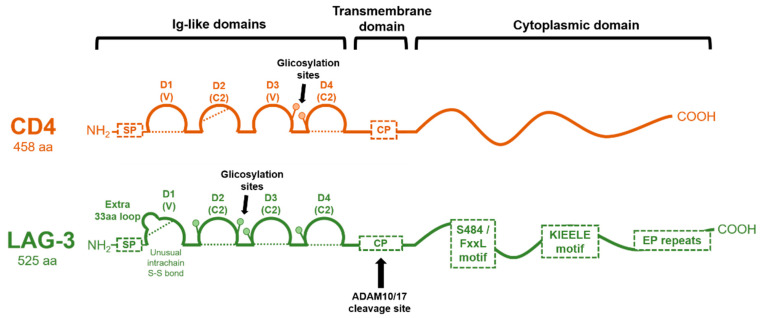
Molecular organization of CD4 and LAG-3 proteins. The domain organization of CD4 and LAG-3 is schematically shown in the figure, with each Ig-like domain indicated as arcs. Dotted lines represent disulfide bonds. The cleavage site for ADAM 10/17 in LAG-3 is shown, rendering a soluble version. CP: connecting peptide; SP: signal peptide.

**Figure 3 ijms-22-05282-f003:**
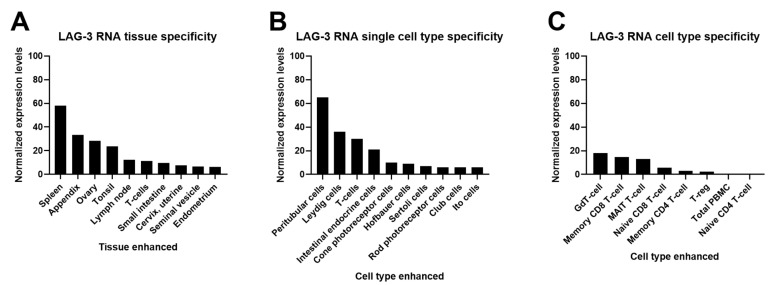
LAG-3 protein and RNA expression profiles from Protein Atlas Analyses (http://www.proteinatlas.org; accessed on 2 May 2021). Images and data credit: Human Protein Atlas. Image and data available from: LAG3 protein expression profiles, The Human Protein Atlas. (**A**) LAG-3 consensus normalized expression (NX) levels for 55 tissue types and 6 blood cell types, created by combining the data from the three transcriptomics datasets (HPA, GTEx, and FANTOM5), using the internal normalization pipeline. Color coding is based on tissue groups by common functional features. RNA tissue specificity is enhanced in lymphoid and ovary tissues. (**B**) Summary of LAG-3 single-cell RNA (NX) from the indicated single cell types. Color coding is based on cell type groups, each consisting of cell types with functional features in common. (**C**) The bar graph represents quantification of RNA-seq data (pTPM) from blood cell types and total peripheral blood mononuclear cells (PBMC) that have been separated into subpopulations by flow sorting.

**Figure 4 ijms-22-05282-f004:**
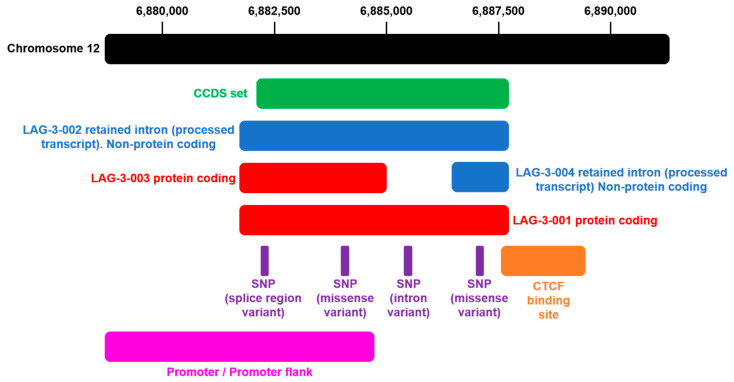
Genomic annotations of the *Lag-*3 gene locus. The *Lag-*3 locus is located at chromosome 12: 6,881,678-6,887,621 forward strand. The illustration shows the organization of the *Lag-*3 locus (http://grch37.ensembl.org/Homo_sapiens/Gene/Summary?db=core;g=ENSG00000089692;r=12:6881678-6887621, accessed on 2 May 2021) [62]. The black bar represents chromosome 12, with positions indicated on top. In green is the *Lag-*3 coding region. In blue is the transcript retaining one intron. In red is the canonical *Lag-*3 transcript. In red and blue is the alternatively-spliced *Lag-*3 transcript. The positions of single-nucleotide polymorphisms are shown below the canonical *Lag-*3 transcript. The promoter regions are also indicated in pink.

**Table 1 ijms-22-05282-t001:** Characterization of human LAG-3 transcripts as listed in ENSEMBLE (https://www.ensembl.org/index.html; accessed on 2 May 2021) [62].

Transcript (ID)	Length	Biotype	Location	Exons	Annotation
*Lag*3-201 (ENST00000203629.3)	Transcript length: 1976 bps Translation length: 525 residues	Protein coding	Chromosome 12: 6,772,520-6,778,455	8 (all coding)	24 domains and features 2174 variant alleles
*Lag*3-202 (ENST00000441671.6)	Transcript length: 1576 bps Translation length: 360 residues	Protein coding	Chromosome 12: 6,772,519-6,775,733	5 (all coding)	19 domains and features 1365 variant alleles
*Lag*3-203 (ENST00000538079.1)	Transcript length: 2587 bps	Retained intron	Chromosome 12: 6,772,512-6,778,455	6 (non-coding)	2174 variant alleles
*Lag*3-204 (ENST00000541049.1)	Transcript length: 684 bps	Retained intron	Chromosome 12: 6,777,450-6,778,455	2 (non-coding)	404 variant alleles

**Table 2 ijms-22-05282-t002:** Clinical landscape of LAG-3-targeted therapy (https://clinicaltrials.gov/, accessed on 2 May 2021.)

Phase	Therapy	NCT Identifier	Intervention/Treatment Tested	Condition or Disease
**Early I**	Monotherapy	NCT04566978	Anti-LAG-3 (89Zr-DFO-REGN3767)	Large B-cell Lymphoma, DLBCL
**I**	Monotherapy	NCT03489369	Anti-LAG-3 (Sym022)	Metastatic Cancer, Solid Tumor, Lymphoma
		NCT03965533	Anti-LAG-3 (GSK2831781)	Healthy Volunteers
		NCT02195349	Anti-LAG-3 (GSK2831781)	Psoriasis
		NCT03538028	Anti-LAG-3 (INCAGN02385)	Select Advanced Malignancies
		NCT00351949	LAG-3-Ig (Eftilagimod Alpha, IMP321)	Metastatic Renal Cell Carcinoma (MRCC)
	Monotherapy and Combination	NCT03252938	LAG-3-Ig (Eftilagimod alpha, IMP321), anti-PD-L1 (Avelumab), standard-of-care chemotherapy	Solid Tumors, Peritoneal Carcinomatosis
		NCT02658981	Anti-LAG-3 (BMS-986016), Anti-PD-1 (Nivolumab, BMS-936558), Anti-CD137 (Urelumab, BMS-663513)	Glioblastoma, Gliosarcoma, Recurrent Brain Neoplasm
		NCT03250832	Anti-LAG-3 (TSR-033), Anti-PD-1	Advanced Solid tumors, Colorectal Cancer
		NCT03005782	Anti-LAG-3 (REGN3767), Anti-PD-1 (Cemiplimab, REGN2810)	Advanced Cancers
		NCT02966548	Anti-LAG-3 (Retalimab, BMS-986016), Anti-PD-1 (Nivolumab, BMS-936558)	Advanced Solid Tumors
		NCT00354263	LAG-3-Ig (Eftilagimod Alpha, IMP321), Agrippal Reference Flu Antigen (commercially available flu vaccine)	Healthy Volunteers
	Combination	NCT04658147	Anti-LAG-3 (Relatlimab), Anti-PD-1 (Nivolumab)	Hepatocellular Carcinoma
		NCT03219268	Anti-PD-1/Anti-LAG-3 DART protein MGD013, Anti-HER2 (Margetuximab, MGAH22)	Advanced Solid Tumors, Hematologic Neoplasms, Gastric Cancer, Ovarian Cancer, Gastroesophageal Cancer, HER2-positive Breast Cancer, HER2-positive Gastric Cancer
		NCT03440437	Anti-LAG-3/PD-L1 Bispecific Antibody (FS118)	Advanced Cancer, Metastatic Cancer
		NCT00732082	LAG-3-Ig (Eftilagimod Alpha, IMP321), Gemcitabine (Gemzar)	Pancreatic Neoplasms
		NCT03742349	Anti-LAG-3 (LAG525, IMP701), Anti-PD-L1 (Spartalizumab, PDR001), Anti-A2AR (NIR178), MET inhibitor (capmatinib, INC280), Anti-M-CSF (MCS110), Anti-IL-1-beta (canakinumab, ACZ885)	Triple Negative Breast Cancer (TNBC)
		NCT04140500	Anti-PD1-LAG-3 Bispecific Antibody (RO7247669)	Solid Tumors
		NCT03849469	Anti-CTLA4-LAG-3 Bispecific Antibody (XmAb^®^22841), Anti-PD-1 (Pembrolizumab (Keytruda^®^))	Selected Advanced Solid Tumors
		NCT03311412	Anti-LAG-3 (Sym022), Anti-TIM-3 (Sym023), Anti-PD-1 (Sym021)	Metastatic Cancer, Solid Tumors, Lymphoma
		NCT02817633	Anti-LAG-3 (TSR-033), Anti-PD-1 (TSR-042), Anti-TIM-3 (TSR-022)	Advanced or Metastatic Solid Tumors
		NCT04252768	LAG-3-Ig (Eftilagimod Alpha, IMP321), Paclitaxel	Metastatic Breast Cancer
		NCT02676869	LAG-3-Ig (Eftilagimod Alpha, IMP321), Anti-PD-1 (Pembrolizumab)	Unresectable or Metastatic Melanoma
		NCT00349934	LAG-3-Ig (Eftilagimod Alpha, IMP321), Paclitaxel	Metastatic Breast Carcinoma
		NCT00354861	LAG-3-Ig (Eftilagimod Alpha, IMP321), hepatitis B antigen (without alum), Engerix B (hepatitis B antigen absorbed on alum)	Healthy Volunteers
		NCT03493932	Anti-LAG-3 (Relatimab, BMS-986016), Anti-PD-1 (Nivolumab)	Recurrent Glioblastoma Patients
		NCT03044613	Anti-LAG-3 (Relatlimab, BMS-986016) Anti-PD-1 (Nivolumab, Opdivo), Carboplatin, Paclitaxel, Radiation	Gastric Cancer, Esophageal Cancer, Gastroesophageal Cancer
		NCT04641871	Anti-LAG-3 (Sym022), Anti-TIM-3 (Sym023), Anti-PD-1 (Sym01)	Advanced Solid Tumor Malignancies
**I/II**	Monotherapy	NCT04618393	Anti-PD-1-LAG-3 Bi-specific Antibody (EMB-02)	Advanced Solid Tumors
	Monotherapy and Combination	NCT04706715	Anti-LAG-3 (89Zr-DFO-REGN3767), Cemiplimab	Metastatic Solid Tumor
		NCT02460224	Anti-LAG-3 (LAG525), Anti-PD1 (PDR001)	Advanced Solid Tumors
		NCT01968109	Anti-LAG-3 (Relatlimab, BMS-986016), Anti-PD-1 (Nivolumab, BMS-936558), (BMS-986213)	Neoplasms by Site, Solid Tumors
	Combination	NCT04611126	Anti-LAG-3 (Relatlimab), Anti-PD-L1 (Ipilimumab), Anti-PD-1 (Nivolumab), Cyclophosphamid, Fludarabine Phosphate, Tumor Infiltrating Lymphocytes infusion	Metastatic Ovarian Cancer, Metastatic Fallopian Tube Cancer, Peritoneal Cancer
		NCT04150965	Anti-LAG-3 (Relatlimab, BMS-986016), Elotuzumab (Empliciti), Pomalidomide, Dexamethasone, Anti-TIGIT (BMS-986207)	Multiple Myeloma, Relapsed Refractory Multiple Myeloma
		NCT01308294	LAG-3-Ig (ImmuFact-IMP321), Tumor Antigenic Peptides (NA-17, MAGE-3.A2, NY-ESO-1, Melan-A, MAGE-A3, MAGE-A3-DP4), Montanide ISA-51	Melanoma
		NCT00365937	LAG-3-Ig (Eftilagimod Alpha, IMP321), Immunological peptides and adjuvants, HLA-A2 peptides (Tyrosinase.A2, MAGE-C2.A2, MAGE-3.A2, MAGE-1.A2, NA17.A2 (GnTV), MAGE-10.A2), Montanide ISA51	Melanoma
		NCT03459222	Anti-LAG-3 (Relatlimab, BMS-986016), Anti-PD-1 (Nivolumab, Opdivo, BMS-936558), IDO1 Inhibitor (BMS-986205), Anti-CTLA-4 (Ipilimumab, Yervoy, BMS-734016)	Advanced Solid Cancers
		NCT02488759	Anti-LAG-3 (Relatlimab, BMS-986016), Anti-PD-1 (Nivolumab), Anti-CTLA4 (Ipilimumab), Anti-CD38 (Daratumumab, Darzalex)	Various Advanced Cancers
		NCT02061761	Anti-LAG-3 (BMS-986016), Anti-PD-1 (Nivolumab, BMS-936558)	Hematologic Neoplasm (Refractory B-Cell Malignancies)
		NCT03610711	Anti-LAG-3 (Relatlimab), Anti-PD-1 (Nivolumab, Optivo)	Gastroesophageal Cancer
		NCT04370704	Anti-PD-1 (NCMGA00012), Anti-LAG-3 (INCAGN02385), Anti-TIM-3 (INCAGN02390)	Melanoma
**II**	Monotherapy	NCT03893565	Anti-LAG-3 (GSK2831781)	Ulcerative Colitis
	Monotherapy and Combination	NCT04080804	Anti-LAG-3 (Relatlimab, BMS-986016), Anti-PD1 (Nivolumab, OPDIVO), Anti-CTLA4 (Ipilimumab, Yervoy)	Head and Neck Squamous Cell Carcinoma (HNSCC)
		NCT03743766	Anti-LAG-3 (Relatlimab, BMS-986016), Anti-PD1 (Nivolumab, BMS-936558)	Melanoma
	Combination	NCT04567615	Anti-LAG-3 (Anti-LAG-3 (Relatlimab, BMS-986016), Anti-PD1 (Nivolumab, BMS-936558)	Advanced Hepatocellular Carcinoma
		NCT03484923	Anti-LAG-3 IgG4 (LAG525), Anti-PD-1 (Spartalizumab, PDR001), Capmatinib (INC280), Canakinumab (ACZ885), Ribociclib (LEE011)	Melanoma
		NCT04634825	PD-1XLAG-3 bispecific DARTmolecule (Tebotelimab, MGD013), Anti-B7-H3 (Enoblituzumab, MGA271), Anti-PD-1 (Retifanlimab, INCMGA00012, MGA012)	Head and Neck Neoplasms
		NCT04326257	Anti-LAG-3 (Relatlimab), Anti-PD-1 (Nivolumab); Anti-PD-L1 (Ipilimumab)	Squamous Cell Carcinoma of the Head and Neck
		NCT03625323	LAG-3-Ig (Eftilagimod Alpha, IMP321), Anti-PD-1 (Pembrolizumab, Keytruda, MK-3475)	Non-Small Cell Lung Carcinoma (NSCLC) and Head and Neck Carcinoma (HNSCC)
		NCT03623854	Anti-LAG-3 (Relatlimab, BMS-986016), Anti-PD-1 (Nivolumab, BMS-936558)	Chordoma
		NCT03662659	Anti-LAG-3 (Relatlimab, BMS-986016), Anti-PD-1 (Nivolumab, Opdivo, BMS-936558), (BMS-986213), XELOX (Oxaliplatin + capecitabine), FOLFOX (Oxaliplatin + leucovorin + fluorouracil), SOX (Oxaliplatin + tegafur/gimeracil/oteracil potassium)	Gastric or Gastroesophageal Junction (GEJ) Cancers
		NCT02614833	LAG-3-Ig (Eftilagimod Alpha, IMP321), Paclitaxel	Adenocarcinoma Breast Stage IV
		NCT03365791	Anti-LAG-3 (LAG525), Anti-PD-1 (PDR001)	Advanced Solid and Hematologic Malignancies
		NCT02060188	Anti-LAG-3 (BMS-986016), Anti-PD-1 (Nivolumab, Opdivo, BMS-936558)	Colorectal Cancer
		NCT02519322	Anti-LAG-3 (Relatlimab, BMS-986016), Anti-PD-1 (Nivolumab, BMS-936558), surgery	Melanoma
		NCT03642067	Anti-LAG-3 (Relatlimab, BMS-986016), Anti-PD-1 (Nivolumab, OPDIVO)	Advanced Colorectal Cancer
		NCT03607890	Anti-LAG-3 (Relatlimab, BMS-986016), Anti-PD-1 (Nivolumab, OPDIVO)	Advanced Mismatch Repair-Deficient Cancers
**II/III**	Combination	NCT04129320	PD-1XLAG-3 bispecific DART protein (MGD013) Anti-B7-H3 (Enoblituzumab, MGA271), anti-PD-1 (MGA012, INCMGA00012)	Head and Neck Cancer
		NCT04082364	Anti-HER2 (margetuximab, MGAH22), Anti-PD-1/anti-LAG-3 dual checkpoint inhibitor DART molecule (MGD013), chemotherapy (XELOX (Capecitabine + Oxaliplatin), mFOLFOX-6 (Leucovorin + 5-FU + Oxaliplatin)	Gastric Cancer, Gastroesophageal Junction Cancer, HER2-positive Gastric Cancer

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
