# Peer review of "Understanding LAG-3 Signaling"

_ijms, 2021, doi:10.3390/ijms22105282_

Round 1
Reviewer 1 Report
This review summarizes the molecular characteristics, expression, and the function in diseases of lymphocyte activation gene 3 (LAG-3). The overall description is relatively complete and has certain reference value for researchers who just learned about LAG-3. Here are some comments/questions:
1) In the abstracts, the authors mentions “Unlike PD-1 and CTLA-4, the exact downstream signaling mechanisms of LAG-3 and its co-signaling with other immune regulatory molecules remains poorly understood”, misleading that this review focus on the poor downstream signaling mechanisms. It might be better put it another way.
2) In Figure 1, what is the principle of distinguishing the three signals? Is signal 2.1 “co-estimulation”? Or “co-stimulation”? What does the black arrow mean, if the green arrow is a positive signal and the red is a negative signal? Why are TCR/CD3 and MHC complexes marked in yellow instead of LAG-3?
3) The format of the line 404 is incorrect.
Author Response
We sincerely appreciate the comments from Reviewer 1, and we thank the Reviewer for the points to improve. We have addressed all the issues as follows, point by point:
1) In the abstracts, the authors mentions “Unlike PD-1 and CTLA-4, the exact downstream signaling mechanisms of LAG-3 and its co-signaling with other immune regulatory molecules remains poorly understood”, misleading that this review focus on the poor downstream signaling mechanisms. It might be better put it another way.
We agree with the Reviewer and we have modified that sentence in the abstract as follows: “Unlike PD-1 and CTLA-4, the exact mechanisms of action of LAG-3 and its relationship with other immune checkpoint molecules remain poorly understood.”
2) In Figure 1, what is the principle of distinguishing the three signals? Is signal 2.1 “co-estimulation”? Or “co-stimulation”? What does the black arrow mean, if the green arrow is a positive signal and the red is a negative signal? Why are TCR/CD3 and MHC complexes marked in yellow instead of LAG-3?
We have corrected the figure and modified the color-coding and the legend of the figure, to make it clearer.
3) The format of the line 404 is incorrect.
We have corrected the format of line 404.
Reviewer 2 Report
In this paper, Chocarro and co-workers provide a review on the mechanisms of LAG-3 signaling and expression regulation, as well as ts participation in several human diseases, from infection, to autoimune and degenerative disorders, to cancer. It finalizes with a tabulation of the main ongoing clinical trials testing LAG-3 targeted therapy.
Overall, the manuscript is well organized and written, with some minor spelling corrections required (see below). Although in the last couple of years some reviews on LAG-3 have been published, the breadth of this review and its timeliness support its interest to the readers of IJMS.
Minor issues:
Line 143 - Should read: After
Lines 199 and 203: there is duplication of information regarding IL-12 up-regulation. Please simplify.
Line 238 - Should read: Hodgkin's
Line 330: Should read: oropharyngeal
Line 331: Should read: urothelial carcinoma (delete "cell")
Line 340: "cancer malignancies" is a pleonasm. Please use "cancers" or "malignancies" as alternatives
Line 365: please delete "pathogenesis"
Line 408: Should read: diabetes mellitus
Conclusions: The authors might further refine this section by providing a critical view on the future developments and potential impact of LAG-3 targeted therapy in human disease, especially cancer.
Author Response
We do appreciate the comments from the Reviewer, whom has been very positive and raised only minor issues. We have corrected all the issues as shown below, point by point:
Minor issues:
Line 143 - Should read: After
Corrected
Lines 199 and 203: there is duplication of information regarding IL-12 up-regulation. Please simplify.
The duplicated “while it is up-regulated by IL-12” part has been removed.
Line 238 - Should read: Hodgkin's
Corrected.
Line 330: Should read: oropharyngeal
Corrected
Line 331: Should read: urothelial carcinoma (delete "cell")
Corrected
Line 340: "cancer malignancies" is a pleonasm. Please use "cancers" or "malignancies" as alternatives
Corrected by “malignancies”.
Line 365: please delete "pathogenesis"
Removed.
Line 408: Should read: diabetes mellitus
Corrected.
Conclusions: The authors might further refine this section by providing a critical view on the future developments and potential impact of LAG-3 targeted therapy in human disease, especially cancer.
We have re-written the conclusion section .